# Cardiac Remodeling According to the Nocturnal Fall of Blood Pressure in Hypertensive Subjects: The Whole Assessment of Cardiac Abnormalities in Non-Dipper Subjects with Arterial Hypertension (Wacanda) Study

**DOI:** 10.3390/jpm11121371

**Published:** 2021-12-15

**Authors:** Domenico Di Raimondo, Gaia Musiari, Alessandra Casuccio, Daniela Colomba, Giuliana Rizzo, Edoardo Pirera, Antonio Pinto, Antonino Tuttolomondo

**Affiliations:** Department of Health Promotion, Mother and Child Care, Internal Medicine and Medical Specialties (ProMISE) “G. D’Alessandro”, University of Palermo, 90100 Palermo, Italy; gaiamusiari@gmail.com (G.M.); alessandra.casuccio@gmail.com (A.C.); daniela.colomba@policlinico.pa.it (D.C.); giulianarizzo@yahoo.it (G.R.); edoardo.pirera95@gmail.com (E.P.); antonio.pinto@unipa.it (A.P.); bruno.tuttolomondo@unipa.it (A.T.)

**Keywords:** blood pressure, essential hypertension, ambulatory blood pressure monitoring, circadian rhythm of blood pressure, hypertensive heart disease

## Abstract

Objective: Several epidemiological studies suggest that the preservation of the physiological circadian rhythm of blood pressure or its disruption affects the extent of the organ damage developed by the patient. If we classify the circadian rhythm of blood pressure into four nocturnal profiles, significant differences emerge in terms of organ damage burden and prognosis: reverse dippers have the worst prognosis while dippers and mild dippers fall into an intermediate risk range. The risk profile of extreme dippers is still debated, and the available data are very conflicting and inconclusive. Starting from this gap of knowledge, we aimed to evaluate, retrospectively, in a cohort of hypertensive subjects, the degree of cardiac involvement in relation to the different nocturnal blood pressure profiles. Methods: We retrospectively evaluated 900 patients with essential hypertension, of whom 510 met our study criteria. We graded the 510 patients in relation to the percentage of reduction in mean systolic blood pressure (SBP) at night-time compared with day-time, considering this as a continuous variable, and then compared the extreme quintiles with each other and with the middle quintile (considered as reference). Results: Patients with less (or no) reduction in nocturnal SBP (reverse dipper) showed a higher level of organ damage and comorbidities. With regard to echocardiographic indexes, patients with maximum nocturnal pressure reduction (extreme dipper) showed a lower level of remodeling and/or impairment of E/e’ ratio, Right Atrium Area, Basal Right Ventricular Diameter, Inferior Vena Cava Average Diameter, and Tricuspidal Anular Plane Systolic Excursion compared also with hypertensive patients with a physiological nocturnal pressure reduction, even after correction for the main confounders. Conclusions: These data suggest that extreme dippers may constitute the subgroup of hypertensive patients with the lowest 24-h pressure load and, therefore, less cardiac remodeling.

## 1. Introduction

Blood pressure (BP) has a physiological circadian rhythm known for many decades [1]. The physiological fluctuation of BP values over 24 h is usually hallmarked by higher levels during day-time and a 10% to 20% fall during night-time. The level of organ damage and the risk of vascular events appear to be significantly influenced by the extent of nocturnal BP reduction [2]. The night-time ambulatory blood pressure (ABP) predicts the outcome of hypertensive patients much better than diurnal. In fact, when diurnal and nocturnal ABP were included in the same models, average nocturnal ABP was superior to diurnal in predicting stroke events as well as all causes and cardiovascular mortality [3,4]. Furthermore, 24-h ABP is a better cardiovascular risk predictor than office BP [5]. These significant amounts of data have driven many authors to recommend a more extensive use of the ambulatory 24-h Blood Pressure monitoring (ABPM) in the clinical/therapeutic management of hypertensive patients [6].

A criticism is related to the evidence that the ABPM predictive value depends on the criterion used to classify the circadian rhythm of blood pressure. By using four nocturnal profiles rather than just two (dippers and non-dippers), significant differences emerge: reverse dippers (patients with an overnight paradoxical increase in BP levels) have the worst prognosis and the higher burden of comorbidities compared to the other three dipping categories [3,7,8,9], while a class of hypertensive patients with still not well-defined prognostic features emerges: extreme dippers. In these subjects there is an exaggerated reduction of nocturnal BP values (>20% vs. mean day-time value); the prognostic relevance of this peculiar characteristic, the risk of vascular events, and the level of organ damage are still highly debated [2].

Studies that have examined ABPM data with the aim of assessing more closely the presence of differences in cardiac involvement according to the nocturnal pressure profile and, therefore, the nocturnal pressure load of the hypertensive patient, have consistently assessed that the non-dipper profile is associated with increased LV-RWT [10,11] and LA enlargement [12], less strongly with an impaired LV diastolic function evaluated through E/A and E/e’ ratio [13], whereas, there is a consistent absence of differences between dippers and non-dippers in left ventricular systolic function assessed by EF [14,15]. Few studies have evaluated the right heart involvement; according to Tadic et al. [16], non-dippers showed an increase in right atrial diameter, a non-statistically significant trend for increased right ventricular wall thickness, whereas TAPSE-estimated right ventricular systolic function did not vary in relation to the nocturnal BP profile. Even more limited is the evidence concerning cardiac involvement comparing four categories of nocturnal dip (revere dipper, non-dipper, dipper, extreme dipper) and not only two (dipper/non-dipper), with inconclusive results especially for extreme dippers.

Starting from this gap of knowledge, we aimed to evaluate retrospectively in a cohort of hypertensive subjects the whole cardiac involvement of extreme dipper hypertensives comparing the level of impairment with the other dipping categories.

## 2. Materials and Methods

We retrospectively evaluated 900 patients affected by essential hypertension among those referring to the outpatient clinic of the Division of Internal Medicine and Stroke Care of the University of Palermo, Italy, in the period between 1 January 2016 and 31 March 2020. Among these, we analyzed those for which all data were available from a transthoracic echocardiogram and an ABPM performed, as well as all anamnestic, clinical, and laboratory data essential to the clinical classification of the patient. 510 patients meet these inclusion criteria.

Exclusion criteria:-Diagnosis of secondary hypertension;-Record of an acute vascular event (ischemic or hemorrhagic stroke, acute myocardial infarction, acute limb ischemia) in the six months prior to the date of the evaluation;-Every condition contraindicating the reliability of the ABPM:
Supraventricular arrhythmias (atrial flutter, paroxysmal, persistent or permanent atrial fibrillation);Clinical history or orthostatic hypotension, autonomic dysfunction or diabetic neuropathy;Body mass Index > 35 Kg/m^2^;History of sleep disturbance (including patients with obstructive sleep apnea) and/or night-workers;

The initial study procedure included the evaluation of the clinical records with special reference to Body Mass Index (BMI) and the following blood biochemical examinations (total cholesterol, HDL cholesterol, triglyceride, creatinine, fibrinogen, complete blood count, fasting glucose, C-reactive protein). Subjects were defined as type 2 diabetics if they had known diabetes treated by diet, oral hypoglycaemic drugs, or insulin.

Previous cerebrovascular disease (TIA/ischemic stroke) was assessed by history, specific neurological examination performed by specialists, and hospital or radiological (brain computed tomography or brain magnetic resonance) records of definite previous stroke.

Previous coronary heart disease (CHD) was assessed by history, clinical examination, electrocardiogram, and echocardiogram.

Essential hypertension was defined on the basis of clinical history, hospital records, and/or antihypertensive treatments taken daily by the individuals. In our study, only subjects with no change in antihypertensive therapy in the last two years have been considered for the analysis. Angiotensin Receptor Blockers (33.1%) and Calcium Channel Blockers (31.5%), was the most commonly used antihypertensive drugs in our sample, followed by ACE-inhibitors (22.4%), beta-blockers (21.9%), diuretics (21.4%), and other drugs (alfa-blockers, clonidine, aliskiren, antialdosteronics) (13.9%).

All enrolled patients had performed ABPM in our unit. The methods of performing and reporting the procedure were as follows. ABPM was performed by a TM-2430 Recorder by A & D Company Limited of Tokyo, Japan. This device provides an oscillometric record. The recorders employed in the current study had previously been validated and recommended for clinical use [17]. The monitoring equipment was arbitrarily applied at 8 AM. The cuff was fixed to the non-dominant arm, and three blood pressure readings were taken concomitantly with sphygmomanometer measurements to ensure that the average of the two sets of values did not differ by >5 mmHg. All patients already taking antihypertensive drugs used their prescribed antihypertensive medications during ABPM, without changes in the type, dosage, or time of administration throughout the study. The device was set to measure blood pressure at 15 min intervals during the day (6:00 a.m. to 10:00 p.m.) and at 30 min intervals during the night (10:00 p.m. to 6:00 a.m.). During the 24 h of examination, the patient was informed to hold the arm immobile at the time of measurements, to keep a diary of daily activities, and to return to the hospital 24 h later. The monitoring was always done on a working day. The patients had no access to the ambulatory BP values. In their diary patients were asked also to indicate the actual time of falling asleep and waking up; if this was significantly different from the preestablished, the calculation of the nocturnal BP reduction was carried out on the actual data.

Measurements recorded during the 24 h were stored on a personal computer and screened as follows: a 24 h record was rejected for analysis if more than one-third of the potential day and night measurements were absent or invalid. The ambulatory BP values used for statistical analysis were expressed as 24-h average systolic and diastolic pressures, and 24-h average heart rate. The night/day ratio of BP was calculated as follows: mean nocturnal systolic BP/mean diurnal systolic BP, considering for the analysis the night/day BP ratio as a continuous variable. According to current guidelines, [18] the four dipper profiles considered for our analysis were identified as follows:-*dippers*: mean reduction in night-time BP between 10 and 20% (night-day BP ratio > 0.8 and ≤0.9; this is commonly considered the physiological profile);-*mild dippers:* night-time drop in BP is between 0 and 10%, i.e., a night-day BP ratio > 0.9 and ≤1);-*reverse dippers:* paradoxical increase in BP during the night (night-day BP ratio > 1);-*extreme dippers*: night-time reduction in BP is >20% (night-day BP ratio ≤ 0.8);

All enrolled patients had performed a transthoracic echocardiogram in our unit. The examinations were performed with a GE-Vivid 7 echocardiographic machine with the patient placed supine and in left lateral decubitus by the same experienced echocardiographer and stored digitally for off-line evaluation using a specific program, to be subsequently interpreted by the same echocardiographer who performed the test. All measurements and echocardiographic data were collected and analyzed in accordance with the latest guidelines of the American Society of Echocardiography (ASE) [19]. The parameters evaluated for the present study are the following:

For the left ventricle:-*Left ventricular end-diastolic volume measured in ml (LV-EDV)*. We considered reference values ranging from 42 and 58.4 mm for men and 37.8 and 52.2 mm for women.-*Left ventricular mass and Ejection Fraction (LVM, EF%)*. The mass of the left ventricle was calculated in grams, using the following formula: 0.8 (1.04 [(LVIDd + IVS + PWT)3/LVID3]) + 0.6 g. where IVS is the end-diastolic interventricular septal thickness, LVID is the left ventricular end-diastolic diameter, and PWT is the posterior wall end-diastolic thickness. The procedure of our study included indexing of left ventricular mass (LVMI), since this system allows the comparison of ventricular masses of subjects having different body weights. We considered a normal left ventricle indexed mass in case of values between 49 and 115 g/m^2^ for men and 43 and 95 g/m^2^ for women. We speak of left ventricular hypertrophy (LVH) when LVMI values exceed 115 g/m^2^ in men and 95 g/m^2^ in women and considered normal a LVEF > of 52% for men and >54% for women.-*Relative Wall Thickness (RWT).*-*Left atrium volume (LAV).* The volume obtained by measuring atrial areas and diameters was indexed for body surface area (LAVi), and a left atrial volume of up to 34 mL/m2 was considered normal in both genders.-*E/A ratio and E/e’ ratio.* LV diastolic function was evaluated through the capture of Tissue Doppler images at mitral cusps level, obtained through a two-dimensional apical window of the four cardiac chambers, were used to measure the transmitral flow velocities; this was measured (in m/s) during the peak of early passive diastolic filling (early, E-wave) and during the late peak of the diastolic flow due to atrial contraction (A-wave). The E/A ratio was then calculated, which decrypts blood flow from the atria to the ventricles during ventricular diastole and provides information about the atrial contribution to ventricular filling. The lower the ratio, the greater the atrial contribution. The E/A ratio value in a subject with normal diastolic function is between 0.8 and 2, but the correct evaluation of the E/A ratio requires a broader framework of the data and, therefore, it is more correct to consider a normal pattern and three abnormal patterns: altered diastolic release (E/A < 1), pseudonormal pattern (E/A falsely in range) and restrictive pattern (E/A > 2).

Pulsed tissue Doppler of the mitral annulus was also used to measure the early protodiastolic peak (e’) of the septal and lateral mitral annulus velocities in addition to the lateral tricuspid annulus velocities; all velocities are expressed in cm/sec. The E/e’ ratio was derived using the mean value of e’ of the septal versant lateral side of the mitral valve. An E/e’ ratio less than 8 is typically associated with normal filling pressures, whereas a ratio > 15 indicates increased filling pressures (ventricular diastolic dysfunction). In accordance with the ASE Recommendations for the Evaluation of Left Ventricular Diastolic Function [20], the final assessment of diastolic dysfunction considered mitral E, E/e’ ratio, and E/A ratio. Using all of these data, a more truthful estimate of the degree to which myocardial release affects the transmitral flow and thus VS filling pressures can be obtained.

For the right ventricle:-*Area of the right atrium.* According to the Guidelines for the echocardiographic assessment of the right heart in adults [21] we considered normal a right atrium area < 18 cm^2^ regardless of sex.-*Basal diameter of the right ventricle.* It was measured at the basal third of the right ventricle and has been considered a normal range between 25–41 mm. The right ventricle is assumed to be dilated if the basal diameter is >41 mm.-*Diameter of the inferior vena cava.* According to the recommendations, a VCI diameter < 21 mm associated with inspiratory collapse > 50% suggests an atrial pressure of 3 mmHg (range 0–5 mmHg), whereas a VCI diameter > 21 mm with inspiratory collapse < 50% suggests an elevated atrial pressure, approximately 15 mm Hg (range 10–20 mmHg). In indeterminate cases, in which the VCI did not meet the two profiles, an intermediate value of 8 mmHg (range 5–10 mmHg) was considered.-*Tricuspid annular plane excursion (TAPSE).* It is measured in M-Mode by placing the cursor on the lateral tricuspid annulus from the apical 4-chamber projection. TAPSE quantifies the systolic excursion of the tricuspid annulus along the longitudinal plane thus assessing the efficiency of contraction; it is, therefore, a reliable index of right ventricular systolic function demonstrating a good correlation with other parameters such as myocardial scintigraphy and 2D estimation of right ventricular ejection fraction. The greater the excursion, the better the performance of the right ventricle. A value > 17 mm was considered normal in both genders.-*Estimated systolic pulmonary artery pressure*. Right ventricular systolic pressure was estimated through the velocity of tricuspid regurgitation using the simplified Bernoulli equation (dP = 4V2), which allows measurement of the right ventricular-atrial pressure gradient. Adding this value to the right atrial pressure estimated by assessing the diameter and respiratory excursions of the inferior vena cava will yield an estimate of pulmonary arterial pressure. The maximum tricuspid regurgitation velocity was considered normal if ≤2.8 m/s, whereas the maximum normal trans-tricuspid gradient was considered for values ≤ 36 mmHg.

### Statistical Analysis

Statistical analysis was performed splitting the population into quintiles with reference to the variable Δ mean SBP Day/Night %, considering it as a continuous variable. The comparison was made between the upper quintile (consisting of the group of extreme dipper subjects), including patients with the most significant nocturnal pressure drop, the lower quintile (consisting of the group of reverse dipper subjects), including patients with less or no nocturnal pressure drop, and the middle quintile, considered the reference quintile (dipper subjects), having a physiological nocturnal reduction in BP. Statistical analysis of quantitative and qualitative data, including descriptive statistics, was used for all items. Continuous variables are presented as mean ± 1 standard deviation where not otherwise specified. The student’s *t* test for paired data was used to compare differences for both continuous anthropometric data and laboratory variables before and after the intervention. Discrete variables were analyzed using the Chi-square test and Fisher’s exact test, if necessary. One way ANOVA test was performed to determine any statistical differences between the means of the quintiles considered. We added a Multivariate Analysis of Covariance (MANCOVA) to correct to correct the differences observed between the quintiles for the major confounder, the mean values of 24 h SBP. A post hoc analysis with the Bonferroni test was added when necessary to confirm the differences between the groups. We also evaluated the strength of the correlation between the nocturnal BP reduction (Δ mean SBP Day/Night %) as a continuous variable and the echocardiographic parameters studied through the calculation of the Pearson’s Correlation Coefficient. Data were analyzed using Epi Info Software (version 6.0, CDC, Atlanta, GA, USA) and SPSS Software version 21.0 (SPSS, Inc., Chicago, IL, USA). *p* values less than 0.05 were considered statistically significant.

## 3. Results

Table 1 shows the demographic and clinical data of hypertensive subjects enrolled according to quintiles of percentage fall in mean nocturnal SBP in comparison to mean diurnal SBP. The quintile that includes the 102 patients with the greatest reduction of nocturnal SBP (extreme dippers, mean nocturnal SBP reduction—22.3%) and the 102 patients with the lowest reduction of nocturnal SBP (reverse dippers, mean nocturnal SBP reduction + 5.5%) have been compared with each other and with the central quintile, considered as reference (mean nocturnal SBP reduction—10.5%, dipper subjects). The three considered quintiles have superimposable mean age, mean duration of hypertension, and mean number and type (see Section 2) of anti-hypertensive drugs taken daily. Compared to other quintiles examined, reverse dipper hypertensives have higher fasting glucose levels: 124.5 ± 42.0 mg/dL vs. 102.8 ± 30.3 and 102.3 ± 33.7 ref and extreme, respectively; *p* < 0.0005; have a higher burden of comorbidities such as diabetes (*p* < 0.0001); impaired renal function (*p* < 0.0005); COPD (*p* < 0.0001); anamnestic vascular events (*p =* 0.033). Extreme dipper hypertensives in our case history have the lowest frequency (compared to all the others, also to the hypertensives with a “normal” circadian rhythm of BP) of diabetes (*p* < 0.0001).

Table 2 shows the ABPM data. Reverse dipper hypertensives besides having the worst nocturnal BP profile have also the highest diurnal SBP levels: 140.0 ± 17.5 mmHg vs. 133.9 ± 11.6 and 131.1 ± 12.1 for ref (*p* = 0.019) and extreme (*p* < 0.0005), respectively; the lowest diurnal DBP levels: 75.8 ± 11.8 mmHg vs. 80.9 ± 8.6 and 83.1 ± 8.6 for ref (*p* = 0.001) and extreme (*p* < 0.0005), respectively. Extreme dipper hypertensives as a result of the increased night-time BP reduction have the highest levels of morning surge for both SBP (+37.4 mmHg, vs. +25.0 and +9.5 for ref and reverse, respectively; *p* < 0.0005) and DBP (+23.3 mmHg, vs. +16.7 and +7.0 for ref and reverse, respectively; *p* < 0.0005). The three quintiles have superimposable diurnal mean SBP and DBP levels.

Table 3 shows the echocardiographic data of hypertensive subjects according to quintiles of percentage fall in mean nocturnal SBP in comparison to mean diurnal SBP. We present both unadjusted and adjusted *p* values for the main confounder mean 24-h SBP value. From the comparison between the top and bottom quintiles with the middle quintile, our data show that reverse dipper hypertensives have a higher level of myocardial function impairment than all the others, whereas extreme dipper hypertensives seem to show merely no index of more cardiac damage than the other quintiles examined but a lower level of cardiac involvement even in comparison with subjects with normal circadian BP rhythm as witnessed by lower LV diastolic impairment (E/e’ 6.75 vs. 8.66; *p* < 0.0005), smaller right atrium size (13.1 vs. 15.2 cm^2^; *p* < 0. 0005) and a generic lesser RV involvement as can be detected by the smaller size of RV basilar diameter (30.1 vs. 31.5; *p* < 0.018, adjusted *p* = 0.05), smaller size of IVC (14.2 vs. 16.3; *p* < 0.0005) and higher values of TAPSE (22.7 vs. 21.1; *p* < 0.0005).

Figure 1, Figure 2, Figure 3 and Figure 4 shows the Pearson’s correlation analysis performed between the nocturnal BP reduction (Δ mean SBP Day/Night %) as a continuous variable and the echocardiographic parameters studied. The analysis showed a linear correlation between Δ mean SBP Day/Night % and LV RWT (r = 0.501, *p* < 0. 0001), LVMI (r = 0.512; *p* < 0.0001), Basal RVD (r = 0.375, *p* < 0.0001) and RA area (0.401, *p* < 0.0001).

Weaker correlations have been found for the other parameters evaluated: LA Volume (r = 0.338, *p* < 0.0001), VCI (r = 0.312, *p* < 0.0001), TAPSE (r = 0.270, *p* < 0.0001), E/A (r = −0.293, *p* < 0.0001).

## 4. Discussion

The main findings of our study are the following:(1)Considering a large cohort of 510 subjects with essential hypertension divided into quintiles with reference to the level of nocturnal BP drop, it is possible to highlight significant differences in the epidemiology, clinic, organ damage, level of comorbidity, and finally cardiac involvement.(2)Mild and reverse dipper hypertensives, as already reported by several authors, are those burdened by a worse clinical, laboratory, instrumental, comorbidity, and organ damage profile.(3)Extreme dipper hypertensives, on the contrary, even when compared with the dipper profile (i.e., compared with subjects with a normal amount of blood pressure reduction during the night) represent the population of subjects with the lowest systemic involvement as well as the lowest level of organ damage.(4)The cardiac involvement during hypertensive disease is different in relation to the nocturnal BP profile, to an extent that seems to go beyond the epidemiological and clinical differences between the groups. A higher nocturnal reduction of SBP seems to be linked to a less marked cardiac involvement;(5)Our data, obtained in groups of subjects with superimposable age, type and mean number of antihypertensive drugs taken daily, duration of hypertensive disease, and day-time BP load, adjusted for the main confounder, 24-h SBP, reaffirms the prognostic relevance of night-time compared with day-time BP.

One of the undoubted advantages of the ABPM use is the possibility to record and use for clinical and therapeutic purposes the measurements obtained during the night-time. If we split hypertensive patients into only two nocturnal dip categories, i.e., dippers (physiological circadian rhythm with a reduction in mean nocturnal SBP > 10%) and non-dippers or mild dippers (reduction in mean nocturnal SBP < 10%), what we observe is higher total mortality, cardiovascular morbidity, and a higher level of organ damage for non-dippers [3,8], but we obtain a much more detailed clinical and prognostic classification of patients if we differentiate the circadian rhythm of BP into four profiles [22,23]. This is because, as underlined by several authors, this approach is most consistent with the actual 24-h trend of BP values [23].

Historically, two patterns of circadian BP variability were associated with an increase in cardiovascular risk: those with higher nocturnal BP levels (reverse dippers and, albeit to a lesser extent, mild dippers) [3,7,8,9,24] and those with a significantly marked reduction in nocturnal BP, linked to an increased risk of cerebral ischemic events (usually occurring in the early morning) and silent cerebral ischemia [25]. It should be noted that an element of “prognostic interference” in extreme dipper patients could be the presence or absence of an exaggerated morning surge, (i.e., the abrupt increase in BP on awakening) that represents itself a risk factor for vascular events. The prognostic variability of extreme dipper subjects reported by various authors should also be evaluated in relation to this co-factor potentially able to mitigate the advantageous effect guaranteed to the extreme dipper patient by the reduced nocturnal and 24-h BP load [26]. To date the prognostic significance of extreme dipping still remains under evaluation: many authors support in their studies that the prognosis of extreme dipper subjects may be superimposable (if not better) than that of dipper hypertensives [2,27,28]. Ben Dov et al. [4] reported in a cohort of 3857 patients a superimposable Hazard Ratio (HR) for all-cause mortality in dipper and extreme dipper after correction for age, sex, hypertension, and antidiabetic treatment. Compared to all dippers taken together, mortality was progressively increasing for mild dippers and reverse dippers (HR of 1.3 and 1.96 respectively). Fagard [22] has reported a better prognosis for extreme dipper compared to the dipper themselves and [23] a statistically significant lower all-cause mortality in extreme dippers than in dippers (*p* < 0.01) with a trend towards the lower frequency of cardiovascular mortality and cardiovascular events (*p*: 0.07). Our data, collected in groups of subjects with superimposable age, type and mean number of antihypertensive drugs taken daily, duration of hypertensive disease, and day-time BP load, seem therefore to support the hypothesis that, for some aspects, such as cardiac organ damage, the extreme dipper BP profile is not to be considered an increased risk profile, but a sort of “*super-dipper*” profile, in which the lower BP load in 24 h due to the greater fall in BP at night can slow down the progression of the disease.

Our data, which show maximal cardiac involvement in subjects with higher nocturnal BP, are in line with literature [13,14,16,24,29]. Left systolic function (assessed by EF%) is not affected in our data by the nocturnal BP profile. This finding seems to support the hypothesis that increased nocturnal BP load does not predominantly cause a higher level of systolic dysfunction, whereas left ventricular diastolic function, assessed as E/A ratio and transmitral E/e’ ratio, seems to be more impaired. These data are also similar to those already reported in the [13,14,16,24,29]. The greater left atrial volume found in subjects with less nocturnal BP reduction also confirms what has already been shown in the literature by other authors [12,16,30,31].

Interesting is our data concerning the different involvement linked to the different nocturnal BP profiles, also of the right sections of the heart, wrongly perceived as less affected by the damage due to the chronic BP increase. The relationship between the LV, which is directly influenced by a persistent overload of central BP, and the RV is extremely complex. The architecture and the pattern of contraction of the two ventricles are intrinsically different. The main mechanisms that make global involvement of the four cardiac chambers in the hypertensive patient pathophysiologically plausible are:(1)Retrograde transmission of the increased afterload through the dysfunctional LV and thus of the dilated LA associated with a possible increase in pulmonary resistance [32]. This mechanism would justify a greater involvement of the right sections in patients with higher mean BP values or in subjects with a highly impaired nocturnal BP fall [33] as we have found in our study.(2)Peculiar susceptibility of the pulmonary circulation of the hypertensive patient to catecholaminergic stress resulting in prolonged vasoconstriction and overload on the RV [34].(3)As a consequence of interventricular septum remodeling induced by chronic BP overload, there was a demonstrated direct mechanical transmission of parietal stress from left to right [35].(4)The activation of bio-humoral mechanisms related to cardiac remodeling in the hypertensive patient (for example the renin-angiotensin-aldosterone system and the atrial natriuretic peptide system) act as much on the left as on the right chambers [33].

Our results are partially comparable with those of Tadic et al. [15], who evaluated the function of the right and left sections of the heart in 376 patients with essential hypertension not under treatment or treated with less than three drugs for less than three years, dividing the population into four categories in relation to the nocturnal BP profile. The authors highlighted how the BP profiles with a higher disruption of the circadian rhythm (non-dipper, reverse dipper) were featured by a worse level of LV and RV diastolic function, without significant differences in systolic function. It should be noted that in this study many of the echocardiographic indices evaluated did not show significant differences by subdividing hypertensives into four dipping groups but only by comparing extreme + dipper vs. reverse + non-dipper. The additional analysis we performed through the calculation of the Pearson’s Correlation Coefficient and shown in Figure 1, Figure 2, Figure 3 and Figure 4 confirms the possibility of identifying a good linear correlation between nocturnal pressure drop and a lower level of myocardial involvement for many of the echocardiographic parameters that we considered. This finding confirms the artifactuality of categorizing a continuous variable such as the ratio of mean day-time SBP to mean night-time SBP and gives strength to our statistical approach. On the other hand, the decision to fix the cut-off of the nocturnal decline of BP assumed to be “normal”, i.e., between 10 and 20%, was arbitrarily defined by a statistical survey in the population [22].

The reasons explaining how a different level of myocardial organ damage can be associated with a different level of nocturnal blood pressure reduction are partly unknown, but they certainly imply multifactorial mechanisms. One determinant is represented by the different sympathovagal balance and the different levels of adrenergic activation. In addition to contributing to the development, maintenance, and progression of the hypertensive state, sympathetic hyperactivity certainly also participates in the genesis of major complications related to target organ damage in the hypertensive patient. It has been reported that in reverse dippers there is sympathetic hyperactivation [36] and reduced sympathetic nervous system fluctuation [37]. The sympathetic hypertone associated with the lack of nocturnal BP reduction is especially marked compared with extreme dippers who seem to have the absolute lowest levels of sympathetic activation [22,38]. Another element that could affect the different involvement of the heart in relation to the nocturnal BP profile could be found in the different nocturnal BP loads. In reverse/mild dippers the physiological nocturnal reduction in cardiac output is lacking, as well as peripheral vascular resistances remain elevated, thus contributing to an increased level of cardiac damage in these patients [39]. An additional element could be the different levels of neurohormonal activation that feature the different dipping profiles. In our study, we did not evaluate neurohormonal parameters such as angiotensin, aldosterone, or NT-proBNP levels, but other authors have reported that a lower nocturnal BP reduction is associated with a relative hyperactivation of the renin-angiotensin system resulting in volume overload [40,41].

Our study has some limitations: the first is the retrospective design of the study; a prospective study designed ad hoc could more accurately define the degree of organ damage in relation to the maintenance over time of the same nocturnal BP profile. The second is the limited reproducibility of nocturnal BP reduction in different monitoring performed in the same subject [18,42]. This issue should make us somewhat cautious in the interpretation and the generalization of our results, although the comparison between such marked averages of night-time blood pressure reduction (−22.3% vs. −10.5%) allows a certain degree of reliability to our analyses.

In conclusion, our study suggests that hypertensive subjects with the greatest reduction in nocturnal BP being the subgroup with the lowest 24-h BP load may have less cardiac remodeling regardless of the duration of the disease, mean 24-h BP levels, and antihypertensive therapy used, reaffirming the prognostic relevance of night-time compared with day-time BP.

## Figures and Tables

**Figure 1 jpm-11-01371-f001:**
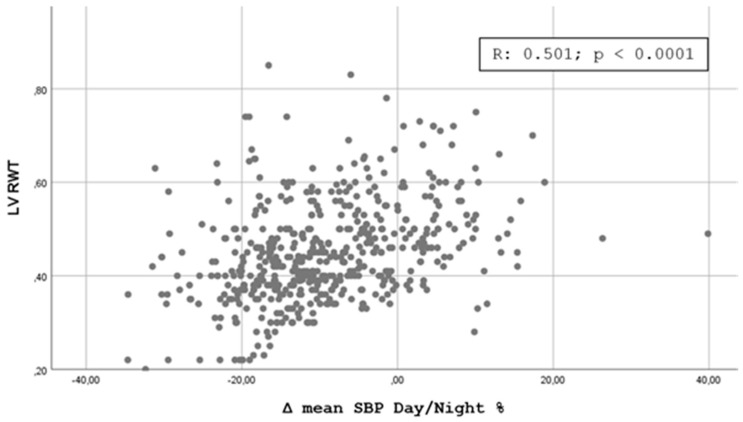
Scatterplot showing the Pearson’s correlation analysis for nocturnal reduction of blood pressure and Left Ventricle RWT.

**Figure 2 jpm-11-01371-f002:**
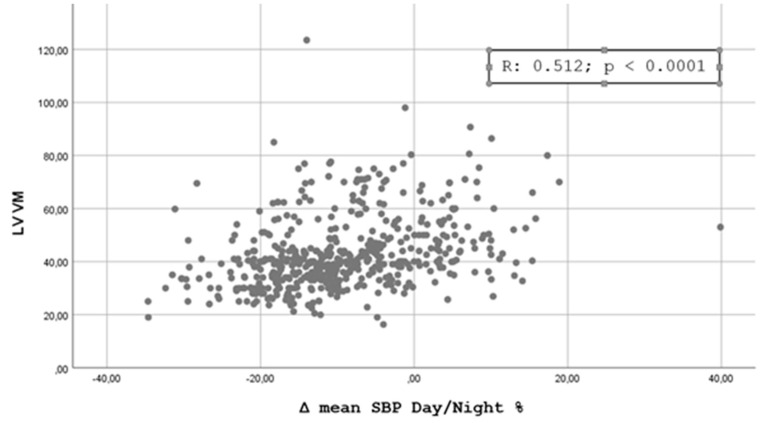
Scatterplot showing the Pearson’s correlation analysis for nocturnal reduction of blood pressure and Left Ventricle Mass Indexed.

**Figure 3 jpm-11-01371-f003:**
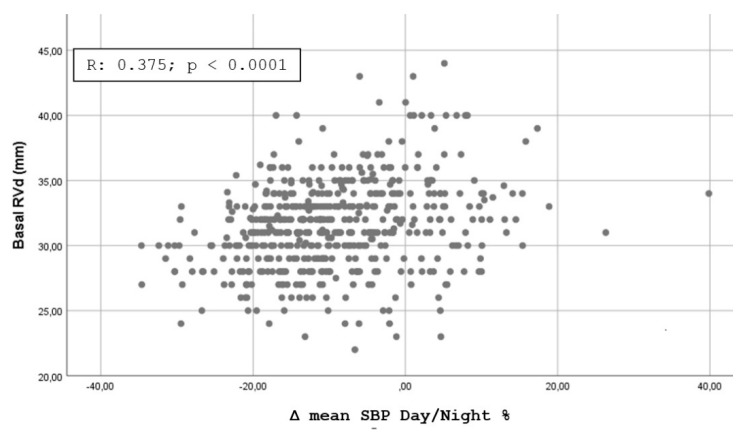
Scatterplot showing the Pearson’s correlation analysis for nocturnal reduction of blood pressure and Basal Right Ventricle Diameter.

**Figure 4 jpm-11-01371-f004:**
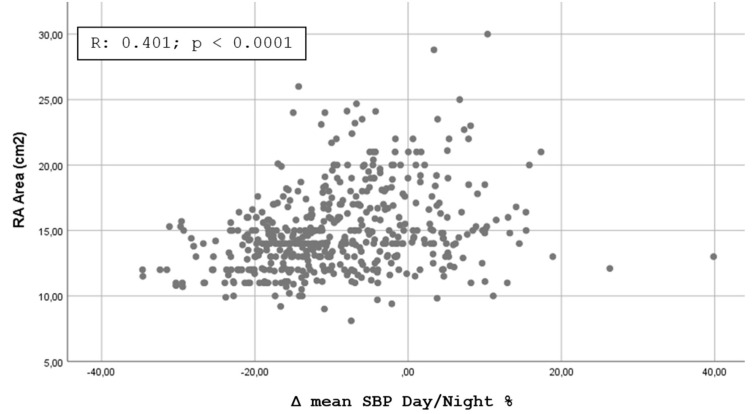
Scatterplot showing the Pearson’s correlation analysis for nocturnal reduction of blood pressure and Right Atrium Area.

**Table 1 jpm-11-01371-t001:** Demographic and clinical data of hypertensive subjects according to quintiles of percentage fall in mean nocturnal systolic blood pressure in comparison to mean diurnal systolic blood pressure.

Variable	Extreme Dipper (n:102)	Dipper (n:102)	Reverse Dipper (n:102)	*p*
M/F, *n* (%)	57/45	57/45	56/46	NS
Mean nocturnal reduction of SBP (%)	−22.3	−10.5	+5.5	0.001
Age (yrs)	66.4 ± 18.3	68.0 ± 14.3	69.7 ± 13.0	NS
Duration of hypertension (yrs)	12.0 ± 9.1	11.6 ± 8.8	12.5 ± 9.3	NS
Family history of hypertension (*n*, %)	70 (68.6)	76 (74.5)	72 (70.6)	NS
Number of drugs taken/day, *n* *	1.89 ± 1.2	2.12 ± 1.2	2.0 ± 1.5	NS
Fasting Glucose (mg/dL)	102.3 ± 33.7	102.8 ± 30.3	124.5 ± 42.0	1 vs. 5 *p* < 0.0005 3 vs. 5 *p* < 0.0005
Oral Hypoglycemic Drugs (*n*, %)	6 (5.9)	2 1 (20.6)	25 (24.5)	1 vs. 3,5 *p* = 0.002
Statins (*n*, %)	26 (25.5)	30 (29.4)	30 (29.4)	NS
BMI (Kg/m^2^)	27.7 ± 4.0	28.5 ± 4.8	29.5 ± 5.7	NS
Waist circonference (cm)	99.5 ±11.9	102.7 ±13.8	103.4 ± 12.9	NS
Total Cholesterol (mg/dL)	200.8 ± 40.3	192.0 ± 42.0	181.3 ± 51.5	1 vs. 5 *p* = 0.022
HDL Cholesterol (mg/dL)	53.1 ± 13.1	49.7 ± 13.4	48.4 ± 15.0	NS
Triglycerides (mg/dL)	115.5 ± 62.2	134.4 ± 65.0	128.0 ± 70.9	NS
Creatinin (mg/dL)	0.97 ± 1.0	0.98 ± 0.68	1.07 ± 0.61	NS
Cr Cl (ml/min) **	102.0 ± 35.8	97.5 ± 39.5	68.8 ± 35.0	1 vs. 5 *p* < 0.0005 3 vs. 5 *p* < 0.0005
MACE (number of events) ***	18	17	35	1, 3 vs. 5 *p* = 0.033
COPD (*n*, %)	3 (2.9)	2 (1.9)	23 (22.5)	1, 3 vs. 5 *p* < 0.0001
Current smokers (*n*, %)	20 (19.6)	16 (15.7)	13 (12.7)	0.290
Past smokers (*n*, %)	28 (27.4)	30 (29.4)	28 (27.4)	0.894
White blood cells count (mm^3^)	7028.0 ± 1855	7313.3 ± 1903	7694.6 ± 2981	NS
hs C-Reactive protein (mg/dL)	0.58 ± 1.5	1.69 ± 6.6	2.21 ±5.4	NS
Fibrinogen (mg/dL)	308.4 ± 81.0	316.9 ± 83.0	338.5 ± 98.1	NS

Table 1—Data are presented as mean value ± DS. BMI means Body Mass Index; Cr Cl: Creatinin Clearance; hsC C-Reactive Protein: High sensitivity C-Reactive Protein; SBP: Systolic Blood Pressure; MACE: Major Adverse Cardiovascular Event; COPD: Chronic Obstructive Pulmonary Disease, NS: Not Significant. * Antihypertensive drugs; the number of molecules and not the number of intakes were considered. ** Calculated applying the formula of Cockroft and Gault. *** MACE identifies the overall number of cardiovascular events (stroke, myocardial infarction or peripheral artery disease requiring hospitalization) in patients’ anamnesis for every quintile.

**Table 2 jpm-11-01371-t002:** ABPM data of hypertensive subjects according to quintiles of percentage fall in mean nocturnal systolic blood pressure in comparison to mean diurnal systolic blood pressure.

Variable	Extreme Dipper (n:102)	Dipper (n:102)	Reverse Dipper (n:102)	*p* *
Mean nocturnal SBP reduction (%)	−22.3	−10.5	+5.5	0.001
24-h SBP (mmHg)	131.1 ± 12.1	133.9 ± 11.6	140.0 ± 17.5	1 vs. 5 *p* < 0.0005 3 vs. 5 *p* = 0.019
24-h DBP (mmHg)	78.0 ± 7.9	78.2 ± 8.4	76.1 ± 11.7	NS
24-h HR (bpm)	74.2 ± 8.7	72.4 ± 8.2	69.9 ± 8.9	1 vs. 5 *p* = 0.007
Day SBP (mmHg)	139.2 ± 13.2	137.8 ± 12.0	137.9 ± 17.5	NS
Day DBP (mmHg)	82.1 ± 8.6	80.9 ± 8.6	78.8 ± 11.8	NS
Day HR (bpm)	77.2 ± 9.5	75.2 ± 8.8	71.4 ± 9.1	1 vs. 5 *p* = 0.001
Night SBP (mmHg)	108.2 ± 10.6	123.4 ± 11.1	145.3 ± 18.5	1 vs. 5 *p* < 0.0005 1 vs. 3 *p* < 0.0005 3 vs. 5 *p* < 0.0005
Night DBP (mmHg)	63.9 ± 7.9	71.1 ± 8.6	76.7 ± 12.1	1 vs. 5 *p* < 0.0005 1 vs. 3 *p* < 0.0005 3 vs. 5 *p* < 0.0005
Night HR (bpm)	65.2 ± 8.5	65.2 ± 7.4	66.0 ± 10.1	NS
Morning surge SBP (mmHg)	+37.4	+25.0	+9.5	1 vs. 5 *p* < 0.0005 1 vs. 3 *p* < 0.0005 3 vs. 5 *p* < 0.0005
Morning surge DBP (mmHg)	+23.3	+16.7	+7.0	1 vs. 5 *p* < 0.0005 1 vs. 3 *p* < 0.0005 3 vs. 5 *p* < 0.0005

Table 2—Data are presented as mean value ± SD. SBP: Systolic Blood Pressure; DBP: Diastolic Blood Pressure; HR: Heart Rate; SD: Standard Deviation, NS: Not Significant. * Post hoc analysis with the Bonferroni test.

**Table 3 jpm-11-01371-t003:** Echocardiographic data of hypertensive subjects according to quintiles of percentage fall in mean nocturnal systolic blood pressure in comparison to mean diurnal systolic blood pressure.

Variable	Extreme Dipper (n:102)	Dipper (n:102)	Reverse Dipper (n:102)	Unadjusted *p*	Adjusted *p* *
LV-EDV (mL)	80.27 ± 23.4	86.8 ± 19.9	91.3 ± 29.5	1 vs. 5 *p* = 0.007	1 vs. 5 *p* = 0.027
LVMi (g/m^2^)	86.9 ± 18.9	95.2 ± 24.3	102.2 ± 49.2	0.506	0.380
LV RWT	0.407 ± 0.116	0.426 ± 0.073	0.510 ± 0.101	1 vs. 5 *p* < 0.0005 3 vs. 5 *p* < 0.0005	1 vs. 5 *p* < 0.0005 3 vs. 5 *p* < 0.0005
LAVi (mL/m^2^)	28.65 ± 7.12	33.18 ± 11.41	40.65 ± 17.74	1 vs. 5 *p* < 0.0005 3 vs. 5 *p* < 0.0005	1 vs. 5 *p* < 0.0005 3 vs. 5 *p* < 0.0005
EF (%)	62.7 ± 5.6	62.5 ± 5.8	60.3 ± 8.4	1 vs. 5 *p* = 0.037	1 vs. 5 *p* = 0.004 3 vs. 5 *p* = 0.012
E/A	1.033 ± 0.369	0.942 ± 0.358	0.733 ± 0.299	1 vs. 5 *p* < 0.0005 3 vs. 5 *p* < 0.0005	1 vs. 5 *p* < 0.0005 3 vs. 5 *p* < 0.0005
E/e’	6.75 ± 2.11	8.66 ± 2.64	12.20 ± 4.36	1 vs. 5 *p* < 0.0005 1 vs. 3 *p* < 0.0005 3 vs. 5 *p* < 0.0005	1 vs. 5 *p* < 0.0005 1 vs. 3 *p* < 0.001 3 vs. 5 *p* < 0.0005
RA Area (cm^2^)	13.1 ± 1.8	15.2 ± 2.8	16.1 ± 3.8	1 vs. 5 *p* < 0.0005 1 vs. 3 *p* < 0.0005	1 vs. 5 *p* < 0.0005 1 vs. 3 *p* < 0.0005
Basal RVD (mm)	30.1 ± 2.4	31.5 ± 2.7	33.0 ± 4.1	1 vs. 5 *p* < 0.0005 1 vs. 3 *p* = 0.018 3 vs. 5 *p* = 0.016	1 vs. 5 *p* < 0.0005 1 vs. 3 *p* = 0.05 3 vs. 5 *p* = 0.032
IVC Diam (mm)	14.2± 2.1	16.3 ± 2.0	16.7 ± 3.7	1 vs. 5 *p* < 0.0005 1 vs. 3 *p* < 0.0005	1 vs. 5 *p* < 0.0005 1 vs. 3 *p* < 0.0005
TAPSE (mm)	22.7 ± 2.7	21.1 ± 2.2	20.4 ± 2.6	1 vs. 5 *p* < 0.0005 1 vs. 3 *p* < 0.0005	1 vs. 5 *p* < 0.0005 1 vs. 3 *p* < 0.0005
PAPS (mmHg)	24.9 ± 4.4	26.7 ± 6.3	28.2 ± 8.4	1 vs. 5 *p* = 0.007	1 vs. 5 *p* = 0.011

Table 3—Data are presented as mean value ± SD. LV-EDV: Left Ventricular End-Diastolic Volume; LVMi: Left Ventricular Mass indexed; LV-RWT: Left ventricular Relative Wall Thickness; LAVi: Left Atrium Volume Indexed; E/e’: E/e’ ratio; RA Area: Right Atrium Area; Basal RVd: Basal Right Ventricular Diameter; IVC Diam: Inferior Vena Cava average Diameter;TAPSE: Tricuspidal Anular Plane Systolic Excursion; APP: Arterial Pulmonary Pressure; LVEF: Left Ventricular Ejection Fraction; E/A: E/A ratio. * *p* adjusted for mean 24-h sistolic blood pressure values. Post hoc analysis with the Bonferroni test.

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
