# Peer review of "Cardiac Remodeling According to the Nocturnal Fall of Blood Pressure in Hypertensive Subjects: The Whole Assessment of Cardiac Abnormalities in Non-Dipper Subjects with Arterial Hypertension (Wacanda) Study"

_jpm, 2021, doi:10.3390/jpm11121371_

Round 1
Reviewer 1 Report
Please consider dividing the studied groups showed in the tables into "dippers", "mild-dippers", etc. instead of quintiles. It would be much more understandable to the readers
Why the non-dippers group is not presented?
Please describe the pharmacological treatment of the studied groups.
Were the patients with the obstructive sleep apnea analyzed?
Please describe the limitations of the study.
Author Response
We would like to thank you for the review of our manuscript. We greatly appreciate the effort you made concerning your critique for the review of our study. We have accepted all your suggestions and revised the article according to them.
Please consider dividing the studied groups showed in the tables into "dippers", "mild-dippers", etc. instead of quintiles. It would be much more understandable to the readers
Thanks for the suggestion. We agree with the reviewer that the tables are easier to read this way. We have modified the tables as suggested. The choice we originally made to indicate the quintiles was motivated by the intention of emphasize the role of considering the variable “Δ mean SBP Day/Night %”, as a continuous variable overcoming the division into categories, which in our opinion can be artificial and in some cases misleading. The explanation of the tables in the text allows us to maintain our original purpose.
Why the non-dippers group is not presented?
Thank you for raising this important point. In many studies assessing the prognosis of hypertensive patients, the non-dipper and reverse dipper groups are often considered together (Ivanovic, B.A.; Tadic, M.V.; Celic, V.P. To dip or not to dip? The unique relationship between different blood pressure patterns and cardiac function and structure. J Hum Hypertens. 2013; 27:62–70. DOI: 10.1038/jhh.2011.83), as they are certainly associated with the worst prognosis compared with all other hypertensives. Having made this premise, our analysis had the main goal to compare extreme dipper hypertensive patients with dipper hypertensive patients (i.e. to compare a group with non-physiological nocturnal BP profile and doubtful prognosis with a group with physiological BP profile and known prognosis) in this perspective the analysis of the reverse dipper group was used only to have a comparison with the group with the absolute worst prognosis. The inclusion in the analysis of another group with known prognosis (relatively unfavorable than the reverse dipper group) would not have added any advantage, in our opinion, also reducing the size of the groups analyzed.
Please describe the pharmacological treatment of the studied groups.
Thanks for the suggestion; we have detailed this information in the materials and methods section.
Were the patients with the obstructive sleep apnea analyzed?
Thank you for the comment. Patients with OSA were excluded, considered in the group of patients with sleep disorders. This has now been specified in the revised version of our manuscript.
Please describe the limitations of the study.
We agree with the reviewer that our study has some limitations that need to be pointed out. We have added the limitations of the study in the final part of the discussion section.
We hope that we have successfully changed our manuscript according to your suggestions and that we have provided all the necessary explanations. We also hope that the manuscript now fulfills your criteria, and the Journal criteria for publication.
Reviewer 2 Report
This study investigates the effect of nocturnal fall of BP on cardiac remodeling assessed by echocardiogram. While this is well-written article presenting intriguing results, several points should be addressed before publication.
1) It seems that the alleged differences in cardiac remodeling between low/middle/high quintiles was ‘unadjusted’ comparison. (Table 3) Though the authors commented that variables such as age, antihypertensive treatments, duration of hypertension and daytime SBP are not different across the quintiles, the average 24-h SBP is significantly different between the quintiles and the differences seem to be large enough to explain the difference in the outcomes. This is a very critical point because adjustment for 24-h SBP might reveal that the relationship between the cardiac remodeling and nocturnal BP reduction is spurious one. Reference # 2 presented results with the adjustment for 24-h SBP.
2) There is a reproducibility concern for the dipping pattern, which may limit the clinical implication of this study. Please refer to articles such as:
Cuspidi, C., Meani, S., Salerno, M. et al. Reproducibility of nocturnal blood pressure fall in early phases of untreated essential hypertension: a prospective observational study. J Hum Hypertens 18, 503–509 (2004). https://doi.org/10.1038/sj.jhh.1001681
Burgos-Alonso, N, Ruiz Arzalluz, MV, Garcia-Alvarez, A, Fernandez-Fernandez de Quincoces, D, Grandes, G. Reproducibility study of nocturnal blood pressure dipping in patients with high cardiovascular risk. J Clin Hypertens. 2021; 23: 1041– 1050. https://doi.org/10.1111/jch.14222
There are many more studies regarding this issue. Thus, it should be commented as a limitation.
3) The authors converted nocturnal BP drop, which is a continuous variable, into a categorical variable. Why not analyze it as it is, especially if the relationship between nocturnal BP drop and cardiac remodeling seems to be linear? Graphic analysis by scatter plot may provide more insight.
4) Did actual wake/sleep time match with the predefined day/night time range?
Author Response
We would like to thank you for your expert review of our manuscript. We greatly appreciate your evaluations regarding our manuscript and the careful suggestions provided that will certainly improve its quality. We put a lot of effort in this study and we appreciate your opinion very much.
This study investigates the effect of nocturnal fall of BP on cardiac remodeling assessed by echocardiogram. While this is well-written article presenting intriguing results, several points should be addressed before publication.
1) It seems that the alleged differences in cardiac remodeling between low/middle/high quintiles was ‘unadjusted’ comparison. (Table 3) Though the authors commented that variables such as age, antihypertensive treatments, duration of hypertension and daytime SBP are not different across the quintiles, the average 24-h SBP is significantly different between the quintiles and the differences seem to be large enough to explain the difference in the outcomes. This is a very critical point because adjustment for 24-h SBP might reveal that the relationship between the cardiac remodeling and nocturnal BP reduction is spurious one. Reference # 2 presented results with the adjustment for 24-h SBP.
Thank you for this valuable comment. Many experimental data show that night-time BP during sleep is closely associated with cardiovascular events and organ damage in hypertensive patients (Kario K. Nocturnal hypertension. Hypertension. 2018;71:997–1009) and that this association is more pronounced and that this association is much stronger than that which exists with mean diurnal blood pressure values (Sega R, et al. Prognostic Value of Ambulatory and Home Blood Pressures Compared With Office Blood Pressure in the General Population Follow-Up Results From the Pressioni Arteriose Monitorate e Loro Associazioni (PAMELA) Study. Circulation. 2005;111:1777–1783). For this reason, having compared groups with identical 24-hour systolic and diastolic blood pressure loads, as shown in Table 2, and on the basis of the large body of supporting data, we considered the differences observed as attributable to the nighttime blood pressure. In the ABC-H Meta-Analysis to which the reviewer refers, published in Hypertension in 2016, Salles GF. et al mainly corrected their data for 24h-SBP, precisely to avoid the possibility that the effects found could be attributable not so much to the different nocturnal fall in BP as to the different 24-hour BP load, but in our case, as already discussed, this confounder is not present. I hope that the reasons for our choice are more understandable now.
2) There is a reproducibility concern for the dipping pattern, which may limit the clinical implication of this study. Please refer to articles such as:
Cuspidi, C., Meani, S., Salerno, M. et al. Reproducibility of nocturnal blood pressure fall in early phases of untreated essential hypertension: a prospective observational study. J Hum Hypertens 18, 503–509 (2004). https://doi.org/10.1038/sj.jhh.1001681
Burgos-Alonso, N, Ruiz Arzalluz, MV, Garcia-Alvarez, A, Fernandez-Fernandez de Quincoces, D, Grandes, G. Reproducibility study of nocturnal blood pressure dipping in patients with high cardiovascular risk. J Clin Hypertens. 2021; 23: 1041– 1050. https://doi.org/10.1111/jch.14222
There are many more studies regarding this issue. Thus, it should be commented as a limitation.
We want to thank the reviewer for raising this important concern. We have added this issue as a limitation of the study in the final part of the discussion section.
3) The authors converted nocturnal BP drop, which is a continuous variable, into a categorical variable. Why not analyze it as it is, especially if the relationship between nocturnal BP drop and cardiac remodeling seems to be linear? Graphic analysis by scatter plot may provide more insight.
Thank you for the comment. Following your suggestion, we evaluated the strength of the association between the nocturnal BP reduction (Δ mean SBP Day/Night %) as a continuous variable and the echocardiographic parameters studied through the calculation of the Pearson's Correlation Coefficient. We found that there is a fairly good and statistically significant level of association for some of the parameters evaluated. We added a scatterplot to the results section showing the main associations identified. We have also edited accordingly the statistical analysis section and the discussion.
4) Did actual wake/sleep time match with the predefined day/night time range?
Thanks for the comment. Patients were asked on the day of monitoring to maintain a routine lifestyle as far as possible, avoiding afternoon naps or going to bed late at night. In any case, patients were asked to indicate the actual time of falling asleep and waking up; if this was significantly different from the preestablished, the calculation of the nocturnal BP reduction was carried out on the actual data. This clarification has now been added to the materials and methods.
We hope that we have successfully changed our manuscript according to your suggestions and that we have provided all the necessary explanations. We also hope that the manuscript now fulfills your criteria, and the Journal criteria for publication.
Round 2
Reviewer 1 Report
Thank you for sending the revised version of the manuskrypt.
The Authors provided all necessary revisions.
I recommend to accept the publication.
Author Response
Thank you for sending the revised version of the manuscript.
The Authors provided all necessary revisions.
I recommend to accept the publication.
We sincerely want to thank the reviewer for the positive evaluation of our work, of which we are very proud.
Reviewer 2 Report
Thanks for the efforts. The revision is reasonable and the manuscript has been significantly improved, except one point (reply to suggested point #1).
In the author's reply, they stated that 'having compared groups with identical 24-hour systolic and diastolic blood pressure loads, as shown in Table 2' confounding by 24hr SBP is not a problem in this case. However, I could not agree with the reply because there are significant differences in 24hr SBP between the groups, which is obvious in the Table 2.
So, still I think multivariable adjustment should be done including 24h SBP to fully support the conclusion.
Author Response
We would like to thank the reviever again for his/her comments and requests for further statistical insights that will certainly allow to strengthen the reliability of our results
Thanks for the efforts. The revision is reasonable and the manuscript has been significantly improved, except one point (reply to suggested point #1).
In the author's reply, they stated that 'having compared groups with identical 24-hour systolic and diastolic blood pressure loads, as shown in Table 2' confounding by 24hr SBP is not a problem in this case. However, I could not agree with the reply because there are significant differences in 24hr SBP between the groups, which is obvious in the Table 2.
So, still I think multivariable adjustment should be done including 24h SBP to fully support the conclusion.
Thanks again for the suggestion. It was not our intent in any way to diminish the value of the comment and the opportunity to give further validity to our findings. We added a Multivariate Analysis of Covariance (MANCOVA) to correct the differences observed between the quintiles for the major confounder, the mean values of 24h SBP. The results are confirmed almost entirely except for small differences that do not change the essence of what our study highlighted regarding the lower cardiac involvement of hypertensive extreme dippers subjects
We hope that we have successfully changed our manuscript according to your suggestions and that we have provided all the necessary explanations. We also hope that the manuscript now fulfills your criteria, and the Journal criteria for publication.